# Molecular basis for the adaptive evolution of environment-sensing by H-NS proteins

Xiaochuan Zhao[1†], Umar F Shahul Hameed[2†], Vladlena Kharchenko[3†], Chenyi Liao[1], Franceline Huser[3], Jacob M Remington[3], Anand K Radhakrishnan[3], Mariusz Jaremko[3], Łukasz Jaremko[3*], Stefan T Arold[2,4*], Jianing Li[1*]

[1]Department of Chemistry, The University of Vermont, Burlington, United States; [2]King Abdullah University of Science and Technology (KAUST), Computational Bioscience Research Center (CBRC), Biological and Environmental Science and Engineering (BESE), Thuwal, Saudi Arabia; [3]King Abdullah University of Science and Technology (KAUST), Biological and Environmental Science and Engineering (BESE), Thuwal, Saudi Arabia; [4]Centre de Biochimie Structurale, CNRS, INSERM, Université de Montpellier, Montpellier, France

**Abstract** The DNA-binding protein H-NS is a pleiotropic gene regulator in gram-negative bacteria. Through its capacity to sense temperature and other environmental factors, H-NS allows pathogens like Salmonella to adapt their gene expression to their presence inside or outside warm-blooded hosts. To investigate how this sensing mechanism may have evolved to fit different bacterial lifestyles, we compared H-NS orthologs from bacteria that infect humans, plants, and insects, and from bacteria that live on a deep-sea hypothermal vent. The combination of biophysical characterization, high-resolution proton-less nuclear magnetic resonance spectroscopy, and molecular simulations revealed, at an atomistic level, how the same general mechanism was adapted to specific habitats and lifestyles. In particular, we demonstrate how environment-sensing characteristics arise from specifically positioned intra- or intermolecular electrostatic interactions. Our integrative approach clarified the exact modus operandi for H-NS-mediated environmental sensing and suggested that this sensing mechanism resulted from the exaptation of an ancestral protein feature.

**\*For correspondence:**
lukasz.jaremko@kaust.edu.sa (ŁJ);
stefan.arold@kaust.edu.sa (STA);
jianing.li@uvm.edu (JL)

[†]These authors contributed equally to this work

**Competing interests:** The authors declare that no competing interests exist.

## Introduction

The histone-like nucleoid-structuring (H-NS) protein is a central controller of the gene regulatory networks in enterobacteria (*Williams and Rimsky, 1997*). H-NS inhibits gene transcription by coating and/or condensing DNA; an environment-sensing mechanism allows H-NS to liberate these DNA regions for gene expression in response to physicochemical changes (*Fang and Rimsky, 2008*; *Winardhi et al., 2015*; *Ali et al., 2012*). H-NS preferentially binds to AT-rich sequences, which enables its dual role in (1) the organization of the bacterial chromosome and (2) the silencing of horizontally acquired foreign DNAs (*Gordon et al., 2011*; *Landick et al., 2015*; *Lang et al., 2007*; *Navarre et al., 2007*). The latter mechanism allows bacteria to assimilate foreign DNAs, which, however, are only expressed as a last resort in case of acute threats or stresses (*Navarre et al., 2007*). Thus, H-NS plays a crucial role in the adaptation, survivability, and antibiotic resistance of bacteria. Given the growing threat of multidrug resistance, H-NS has attracted increasing research interest, with a particular focus on elucidating the molecular mechanisms of adaptive evolution (*Ali et al., 2014*; *Will et al., 2015*; *Higashi et al., 2016*).

H-NS possesses two dimerization domains (site1, residues 1–44; site2, resides 52–82; the numbering of *Salmonella typhimurium* is adopted throughout the text) and a C-terminal DNA-binding domain (DNAbd, residues 93–137) that is connected through a flexible region (linker, residues 83–

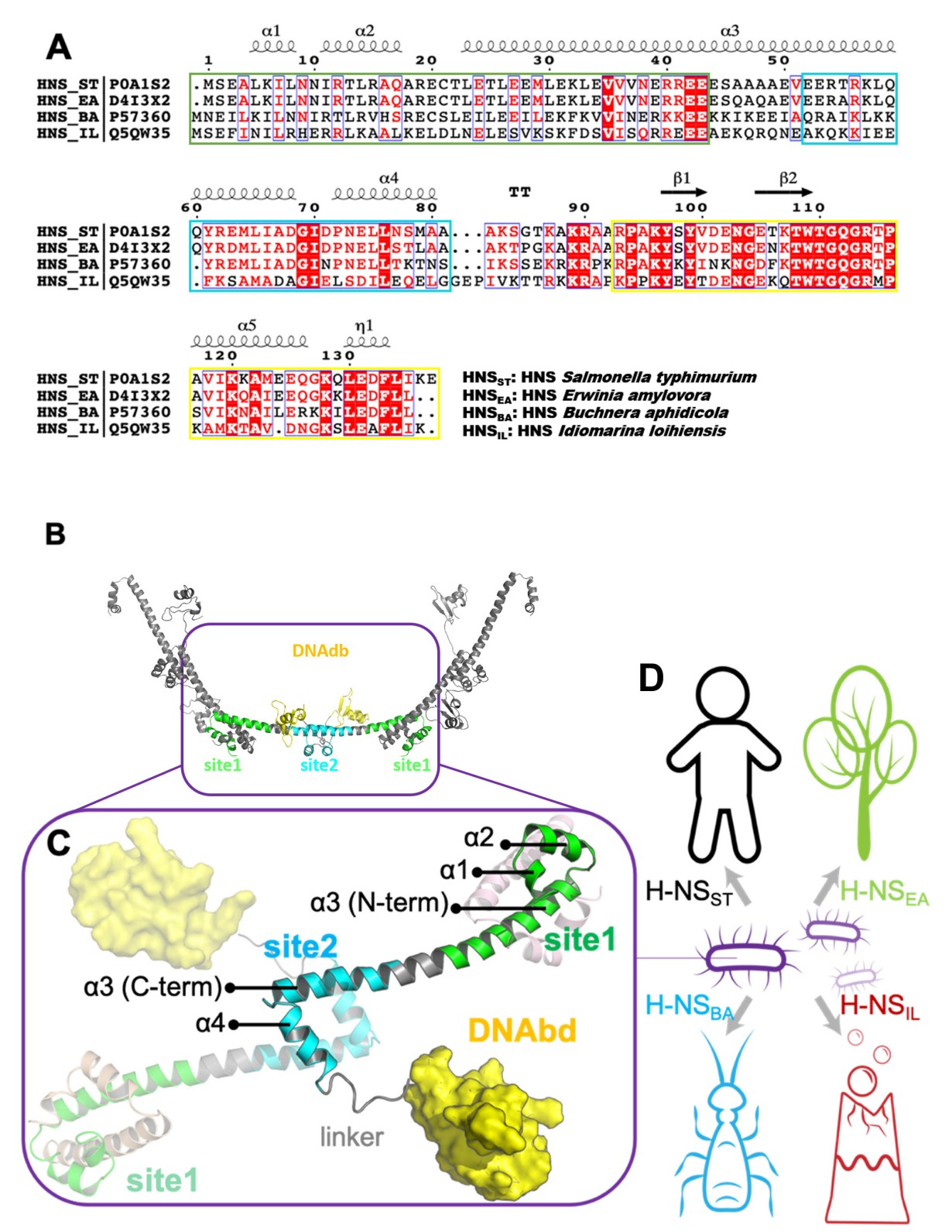

**Figure 1.** The sequence, structure, and habitat of selected histone-like nucleoid-structuring (H-NS) orthologs. (**A**) Sequence alignment of H-NS orthologs (prepared with ESPript 3.0) (*Robert and Gouet, 2014*): H-NS$_{ST}$ (*Salmonella typhimurium*; UniprotID: P0A1S2), H-NS$_{EA}$ (*Erwinia amylovora*; UniprotID: D4I3 × 2), H-NS$_{BA}$ (*Buchnera aphidicola*; UniprotID: P57360), and H-NS$_{IL}$ (*Idiomarina loihiensis* UniprotID: Q5QW35). The color scheme for sequence similarity is as follows: red background (identical in four orthologs) > red font (similar residues) > black font (totally different). Green, cyan,
*Figure 1 continued on next page*

*Figure 1 continued*

and yellow frames indicate the site1 (residues 1–44), site2 (resides 52–82), and DNAbd (residues 93–137). (B) Multimer formed by H-NS$_{ST}$. The model was built based on the available experimental structures (PDB accessions 3NR7 and 2L93). One H-NS$_{ST}$ dimer shows site1, site2, and DNAbd colored as in (A) (green, cyan, and yellow, respectively). (C) The tetrameric H-NS$_{ST}$ model we used for molecular dynamics (MD) is shown. The central dimer has each domain color-coded as in (A) and (B). (D) Illustration of the environment of the selected orthologs.

92) to site2 (*Figure 1*; *Gordon et al., 2011*; *Shindo et al., 1995*; *Bloch et al., 2003*; *Arold et al., 2010*; *Gao et al., 2017*). The combination of site1 'head-to-head' dimers with site2 'tail-to-tail' dimers allows H-NS to multimerize (*Figure 1B*). These H-NS multimers form a superhelix that recapitulates the structure of plectonemic DNA, offering a mechanism for a stable concerted DNA coating by H-NS that results in gene silencing (*Arold et al., 2010*). However, other modes of DNA association by H-NS were also proposed (*Qin et al., 2019*).

In a previous study, we showed that site2 of *S. typhimurium* H-NS is the primary response element to temperature changes (*Shahul Hameed et al., 2019*). Site2 unfolds at human body temperature, allowing the linker-DNAbd region to associate with site1 to adapt an autoinhibited conformation incapable of binding to DNA. Salinity and pH can also influence the stability of site2 dimers and hence may also affect gene repression by H-NS (*Shahul Hameed et al., 2019*; *van der Valk et al., 2017*). Thus, the sensitivity of H-NS to temperature and other physiochemical changes allows human pathogens such as *S. typhimurium*, *Vibrio cholerae*, and enterohaemorrhagic *Escherichia coli* to sense when they enter a homothermic host and adapt their gene expression profiles accordingly.

To date, studies to elucidate environment-sensing of H-NS were almost exclusively conducted with proteins from two model systems, *S. typhimurium* (e.g. *Ali et al., 2014*; *Navarre et al., 2006*; *Ali et al., 2013*; *Hu et al., 2019*) and *E. coli* (e.g. *van der Valk et al., 2017*; *Oshima et al., 2006*; *White-Ziegler and Davis, 2009*; *Kahramanoglou et al., 2011*; *Ueda et al., 2013*; *Kotlajich et al., 2015*), both of which infect humans. Yet, H-NS orthologs are also present in enterobacteria that do not have warm-blooded hosts, raising the question of what biological role H-NS plays in these species. Answering this question requires to determine the structural basis for environment-sensing in H-NS orthologs with drastically different lifestyles. However, the multidomain composition of H-NS hamper conventional structural analysis. Therefore, we combined large-scale molecular simulations and spectroscopic approaches to elucidate how environment-sensing by H-NS may have adapted in different species. This multidisciplinary approach yielded an atomic-level understanding of how H-NS orthologs evolved specific residue substitutions to adapt environment-sensing to their bacterial habitats, and may open new avenues for strategies to combat antibiotic resistance.

## Results

To investigate the adaptation of environment-sensing by H-NS, we searched for representatives of H-NS–containing bacteria that have diverse lifestyles. Accordingly, we selected four H-NS orthologs from ~3000 hr-NS-like sequences available in the Uniprot database: (1) H-NS$_{ST}$ from *S. typhimurium*. This bacterium is a pathogen of mammals and uses temperature-sensing to adapt to a presence inside the warm-blooded host. (2) H-NS$_{EA}$ from *Erwinia amylovora*, which is a plant pathogen that infects apples and pears. Hence, temperature is not a reliable differentiator between free-living and host-based states. (3) H-NS$_{BA}$ from *Buchnera aphidicola*. This bacterium is an obligate endosymbiont of aphids and has no free-living forms. (4) H-NS$_{IL}$ from *Idiomarina loihiensis*, which is a free-living bacterium from a deep-sea hydrothermal vent producing large heat gradients. H-NS$_{EA}$ and H-NS$_{BA}$ share more than 60% sequence identity with H-NS$_{ST}$, whereas H-NS$_{IL}$ is only 40% identical to H-NS$_{ST}$ (*Figure 1A*, *Supplementary file 1A*).

### The site1 dimer is markedly more stable than the site2 dimer in the H-NS orthologs

H-NS$_{ST}$ site1 and site2 form homodimers to enable H-NS multimerization in a head-to-head/tail-to-tail fashion (*Figure 1B*; *Arold et al., 2010*). In concert with the DNA interaction of the individual domains, this homo-oligomerization is required for tight DNA binding and hence gene repression. In our previous study, we showed that only H-NS$_{ST}$ site2 dimers unfold and dissociate within a

biologically relevant temperature range, whereas site1 dimers remain unaffected (*Shahul Hameed et al., 2019*). The higher stability of the site1 dimer of H-NS$_{ST}$ is explained by a substantially larger contact surface between the two monomers (ca. 3300 Å$^2$ compared to ca. 850 Å$^2$ for site1 and site2, respectively, according to PDBePISA [*Krissinel and Henrick, 2007*]).

To investigate whether this mechanism is conserved in other H-NS orthologs, we built homology models for H-NS$_{EA}$, H-NS$_{BA}$, and H-NS$_{IL}$ using the crystal structure of the H-NS$_{ST}$ site1–site2 fragment as a template (PDB ID: 3NR7) (*Arold et al., 2010*). Next, we constructed a tetrameric model as a minimal representation that conserves all features of the H-NS multimer. This tetramer contained two full-length H-NS monomers (residues 1–137, with templates PDB IDs: 3NR7 and 2L93) and two partial monomers, truncated before site2 (residues 1–52) (*Figure 1C*). To probe differences in environmental responses of the orthologs, we first used conventional all-atom MD. We simulated all four tetramers (~100,000 atoms in each system; see Materials and methods) for 200 ns at three different conditions (0.15 M NaCl, 293 K; 0.50 M NaCl, 293 K/20°C; or 0.15 M NaCl, 313 K/40 °C) (*Supplementary file 1B*).

The tetramer simulations at 0.15 M NaCl and 20°C produced a lower residue fluctuation level in site1 (local root-mean-square fluctuation [RMSF] 0.4–1.9 Å) than in site2 (local RMSF 0.5–4.4 Å) for all four orthologs (*Supplementary file 1C*). The higher stability of the site1 dimer is explained by the generally higher number of nonpolar contacts than in the site2 dimer (*Figure 2—figure supplement 1*). These contacts involved conserved hydrophobic amino acid residues, notably L5 (or I5) and L8 of α1, L14 of α2, and L23, L26, V36, and V37 (or I37) of α3 (*Figure 2*). Our in silico mutant stability prediction analysis corroborated qualitatively the importance of hydrophobic residues for stabilizing the site1 dimer, in particular of L5, L8, L23, and L26 (*Supplementary file 1D and 1E*).

These interactions remained formed in all site1 dimers in our tetramer simulations (at 0.15 M NaCl at 20°C) and tetramer simulations at higher salinity (at 0.50 M NaCl at 20°C) or higher temperature (at 0.15 M NaCl at 40°C). Hence, we found that the stability of the site1 dimers resulted mainly from strong and conserved nonpolar packing. Indeed, recombinantly expressed site1 fragments of all four orthologs formed ~15 kDa dimers in size exclusion chromatography–multi-angle light scattering. These dimers were stable within the temperature range relevant for environment-sensing (aggregation temperature, $T_{agg}$ > 37 °C; *Figure 2—figure supplement 2*). We concluded that the mechanism observed for H-NS$_{ST}$ – where site1 remains stable and the site2 stability is affected by the environment – is conserved in H-NS$_{EA}$, H-NS$_{BA}$, and H-NS$_{IL}$.

## Variations in the site2 sequence alter the sensing sensitivity of H-NS orthologs

Compared to site1 dimers, site2 dimers harbor fewer nonpolar contacts, only involving residues L58 (or I58), Y61 (or F61), M64 (or A64), I70, and L75 (or I75) (*Figure 2*). Hence, while site1 dimerization was largely maintained by nonpolar packing, site2 dimerization was strongly driven by electrostatic interactions from selective salt bridges. MD simulations revealed that these salt bridges were in a dynamic equilibrium between forming, breaking, and rearranging. These salt bridges were either formed in cis, within the site2 monomer, (e.g. E52-R56 and R62-E63 in H-NS$_{ST}$) or in trans, between two monomers in the site2 dimer (e.g. R54-D71', R54-E74', and K57-D68' in H-NS$_{ST}$, where the apostrophe denotes residues from the second chain; illustrated in *Figure 3*). In addition to substitutions that delete (E52A in H-NS$_{BA}$; E63S and D68A in H-NS$_{IL}$) or weaken (E63Q in H-NS$_{EA}$; D71N in H-NS$_{BA}$) these salt bridges, our simulations showed different levels of site2 salt bridge stability among orthologs (*Figure 3*, *Supplementary file 1F*): (1) The inter-monomer salt bridge R/K54-E/D74' was stable in all our simulations at 20°C and 0.15 M NaCl, but less likely to form at an increased temperature (40°C) or salinity (0.50 M NaCl), suggesting that this salt bridge is involved in environmental sensing (*Figure 3A*). (2) Absent in H-NS$_{IL}$, the inter-monomer salt bridge K57-D68' remained formed during all our simulations of H-NS$_{ST}$, H-NS$_{EA}$, and H-NS$_{BA}$, indicating a 'housekeeping' role for the stability of the site2 dimer in all orthologs except for H-NS$_{IL}$ (*Figure 3B*).

Our simulations show how specific protein dynamics might modulate the ortholog's response to salinity or temperature. For example, we observed increased bending of the α3 backbone (annotated by the black arrow in *Figure 3C*) at high temperature (40°C) or high salinity (0.50 M NaCl) (*Figure 3—figure supplement 1*). Although α3 bending occurred in all orthologs, it only significantly affected the site2 dimer of H-NS$_{ST}$ by separating R54 from E74' or D71', suggesting that this

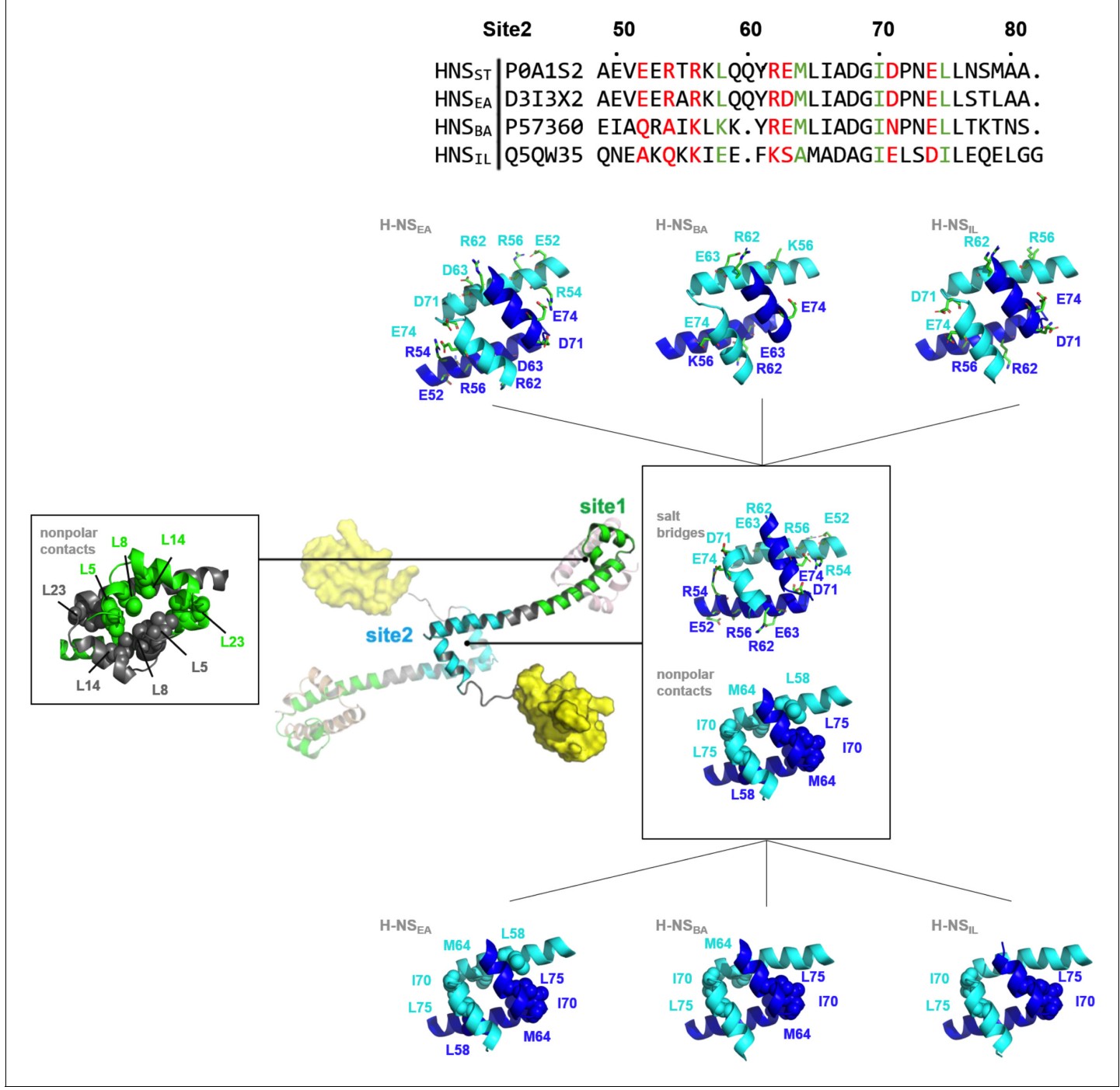

**Figure 2.** Nonpolar and ionic interactions in histone-like nucleoid-structuring (H-NS) site1 and site2 dimers. Hydrophobic contact residues are shown as sphere models and polar contact residues as stick models. The two protein chains forming the dimer are color-coded. The site2 sequence alignment indicates in red the residue sites that potentially form salt bridges in H-NS$_{ST}$, and in green the residues that form nonpolar contacts in H-NS$_{ST}$. For additional details, see Figure S1.

The online version of this article includes the following figure supplement(s) for figure 2:

**Figure supplement 1.** Computational analysis of H-NS nonpolar contacts and site2 dynamics.

**Figure supplement 2.** Dimerization and stability of H-NS 1-57.

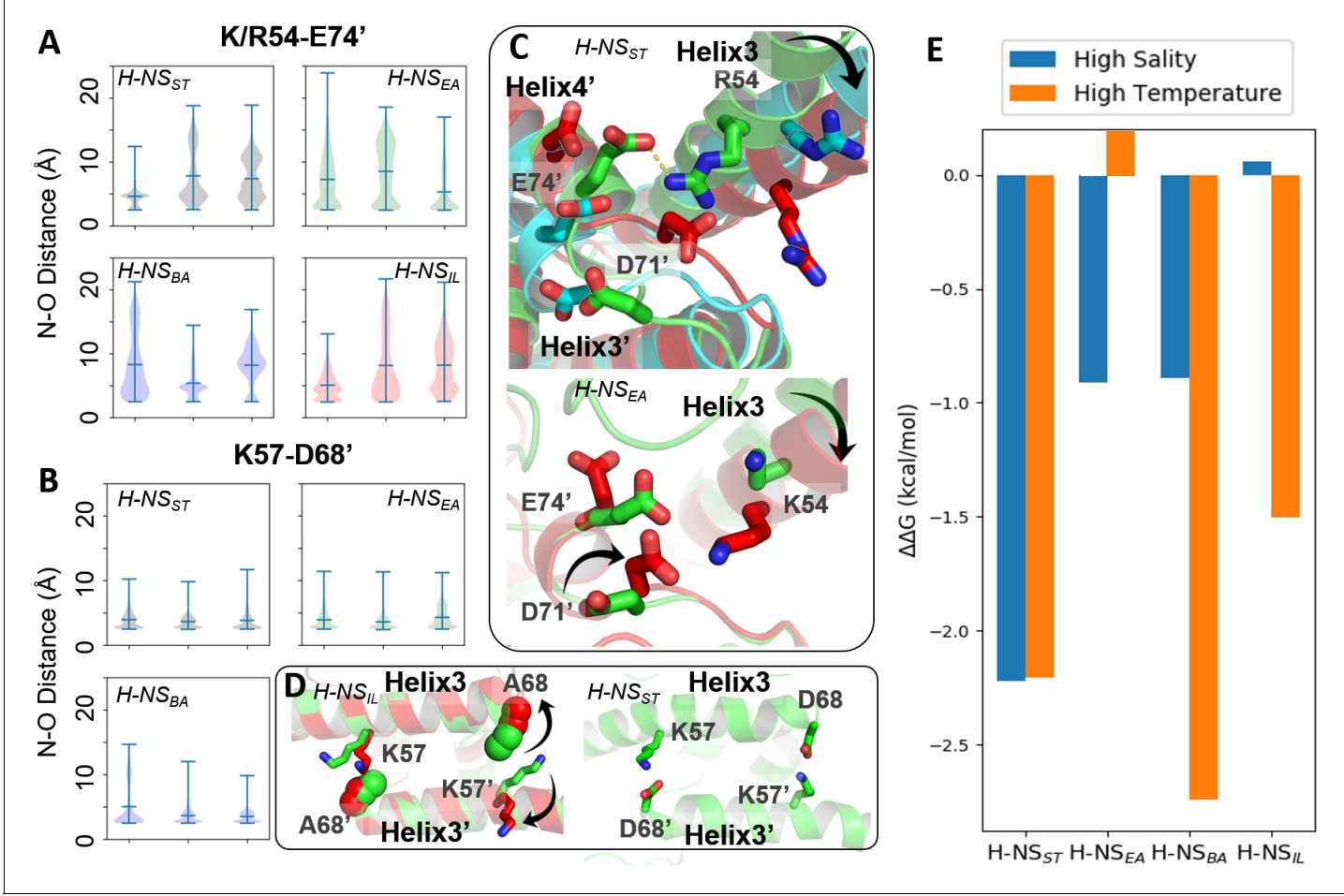

**Figure 3.** Conserved inter-dimer salt bridges observed in molecular dynamics (MD) simulations. (A, B) Violin plots of the distance between the nearest side chain nitrogen atom of lysine/arginine and the side chain oxygen atom of aspartic/glutamic acid in the salt bridge. Each subplot shows the results obtained at 20°C, 0.15 M NaCl (left), 40°C, 0.15 M NaCl (middle), and 20°C, 0.50 M NaCl (right). Each 'violin' displays the mean value (the bar in the center of the violin), the range (the stretched line), and the distribution of the distance (kernel density on the side). As we use the numbering of H-NS$_{ST}$, there are position shifts in H-NS$_{BA}$ and H-NS$_{IL}$: R54 to R53, K57 to K56, and D68 to D67 in H-NS$_{BA}$; R54 to K53 in H-NS$_{IL}$. (C) Final snapshots of the R54-E74' salt bridge in H-NS$_{ST}$ and K54-D71' in H-NS$_{EA}$. Color scheme of the cartoon: 20°C, 0.15 M NaCl (green), 40°C, 0.15 M NaCl (red), and 20°C, 0.50 M NaCl (cyan). (D) Final snapshots of the K57-D/A68 contact in H-NS$_{ST}$ and H-NS$_{IL}$. Same color scheme as (C). (E) Free energy changes as a result of increased salinity or temperature, according to the potential of mean force (PMF) calculated from umbrella sampling.

The online version of this article includes the following figure supplement(s) for figure 3:

**Figure supplement 1.** Representative H-NS monomers (with site2 alignment) from the final snapshots (at 200 ns) of the tetramer simulations under different conditions: 20°C, 0.15 M NaCl (green), 20°C, 0.50 M NaCl (cyan), and 40°C, 0.15 M NaCl (red).

**Figure supplement 2.** Umbrella sampling for H-NS site2 dimers.

mechanism contributed to the salt and temperature sensitivity of H-NS$_{ST}$ site2, whereas it was not strong enough to significantly affect site2 stability in other orthologs.

Another example was given by H-NS$_{EA}$, where an alternative R54-D71' salt bridge formed whenever the R54-E74' contact was broken at 40°C. This alternative R54-D71' salt bridge stabilized the H-NS$_{EA}$ site2 dimer at the higher temperature, suggesting that this compensatory mechanism resulted in a decreased sensitivity to temperature (*Figure 3C*). H-NS$_{IL}$ provided a final example for a specific response. Compared with the R54-E74' salt bridge (*Figure 3A*), the K57-D68' salt bridge only varied slightly in all our simulations (*Figure 3B*). However, the substitution D68A in H-NS$_{IL}$ supplanted the electrostatic interaction with a nonpolar interaction, which was broken at 40°C in our simulations (*Figure 3D*). This effect suggested that H-NS$_{IL}$ had a reduced sensitivity to salinity, while remaining sensitive to temperature.

To complement the dynamics of H-NS orthologs from our conventional MD simulations, we used extensive simulations with umbrella sampling (US) to quantitate the overall site2 stability. We calculated the potential of mean force (PMF) for site2 dimer dissociation (residues 50–82, ~46,000 atoms) of the four H-NS orthologs for three different conditions (low salinity/low temperature, high salinity, or high temperature). The site2 monomers were not constrained and remained structurally flexible during the dissociation process. To ensure convergence in the PMFs, we employed long windows (54 ns) in simulations totaling 52 μs (details provided in the SI; see *Figure 3—figure supplement 2* for resulting histograms and PMFs along the dissociation coordinate). According to the free-energy difference between the dimerization and dissociation states ($\Delta G = G_{dimer} - G_{dissociation}$), we estimated the energetic impact from increased salinity and temperature as follows: $\Delta\Delta G = \Delta G_{high\ salinity\ or\ temperature} - \Delta G_{293K,\ 0.15M\ NaCl}$ (*Figure 3E*). Notably, high salinity (0.50 M NaCl) or temperature (40° C) decreased the stability of the H-NS$_{ST}$ site2 dimer by 2.2 kcal/mol. H-NS$_{BA}$ displayed a similar sensitivity to temperature but a lower sensitivity to salinity, which destabilized the dimer by 1.5 kcal/mol. Interestingly, our data indicated that H-NS$_{EA}$ was only sensitive to salinity, whereas raising the temperature had little impact on the stability of the H-NS$_{EA}$ site2 dimer. Conversely, H-NS$_{IL}$ only responded to temperature, whereas the increased salinity did not affect the stability of its site2 dimer ($\Delta G \sim 0$ kcal/mol). Collectively, our conventional MD simulations and PMF calculations suggested how, on the atomic level, changes in the site2 sequence may alter the sensitivity of the H-NS orthologs to different environmental changes.

## The autoinhibited H-NS conformation is maintained through dynamic electrostatic interactions

In a previous study (*Shahul Hameed et al., 2019*), we had shown that melting and dissociation of site2 dimers allow H-NS$_{ST}$ to adapt a closed conformation in which the linker-DNAbd fragment interacts with a negatively charged region on site1 α3 (*Figure 4—figure supplement 1A*) and that this autoinhibitory interaction is incompatible with DNA interactions. However, due to extensive signal broadening of mainly linker amides exchanging with water, our conventional proton-detected NMR analysis based on exchangeable amide H/N-observed correlations did not allow confident mapping of the binding site on the C-terminal region (*Shahul Hameed et al., 2019*; *Figure 4—figure supplement 1B*). Herein, we overcame this limitation by using proton-less $^{13}$C-detected NMR analysis to complete the resonance assignment of the linker-DNAbd fragment (*Figure 4* and *Figure 4—figure supplement 1C*). These complete carbon chemical shifts allowed us to elucidate the structural mechanism of H-NS$_{ST}$ autoinhibition fully and, in a second step, to use this understanding to investigate the existence of this closed conformation in the orthologs.

We first titrated unlabeled H-NS$_{ST}$ site1 (residues 1–57) onto the $^{13}$C,$^{15}$N-labeled H-NS$_{ST}$ C-terminal region (Ct$_{ST}$, residues 84–137), comprising the linker (residues 84–93) and DNAbd (residues 94–137) (*Figure 4A*, *Figure 4—figure supplement 1A*). Motif identification from chemical shifts (MICS) (*Shen and Bax, 2013*) revealed that the interaction promoted the formation of a short type VIII β-turn in residues 89–92 (MICS confidence coefficient was 0.69). No significantly stable other motif besides the residual random coil was identified (*Figure 4B,C*). This sharp turn brings the positively charged linker side chains K87, K89, R90, R93 closer to each other than in the free state, presumably as a result of pairing them with opposite charges on site1 (*Shahul Hameed et al., 2019*; *Figure 4B, C*).

To further probe the local dynamics of the polypeptide chain, we determined the random-coil-index order parameter RCI-$S^2$ based on the fully assigned $^{13}$C-resonances for each residue for the ligand-free and site1-saturated Ct$_{ST}$. The dynamics of the well-ordered DNAbd domain remained unchanged with or without site1 present, in agreement with its only minor involvement in the autoassociation (*Figure 4D*). Conversely, the linker residues 84–95 were disordered without regular secondary motifs in the absence of site1 (RCI-$S^2$ < 0.35). Upon addition of site1, the local dynamics decreased, particularly within the stretch of four amino acids K89-R90-A91-A92 (RCI-$S^2$ > 0.6) that predominantly form the type VIII β-turn according to MICS. Nonetheless, the overall RCI-$S^2$ of the linker remained low, although experimental conditions resulted in >99% of ligand saturation of the labeled Ct$_{ST}$, demonstrating that the association with site1 did not substantially restrict the linker's movements (*Figure 4D*).

Collectively, our analysis established that the autoinhibitory site1:Ct$_{ST}$ association was driven by oppositely charged residues located on site1 and the linker, and involved only a small region of the

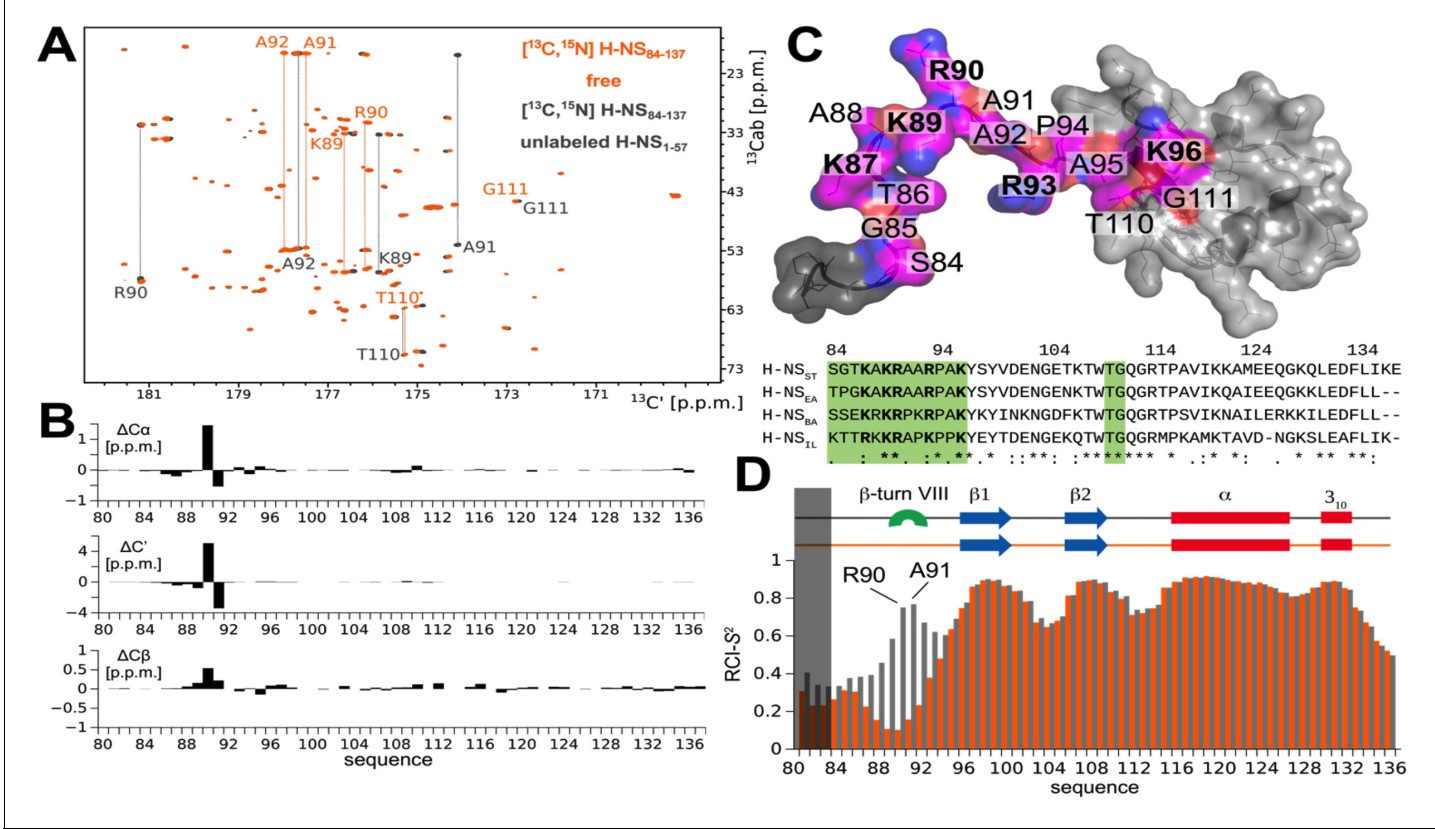

**Figure 4.** The atomistic details of histone-like nucleoid-structuring (H-NS) autoinhibition revealed by high-resolution proton-less low-γ detected NMR. (A) The 2D CBCACO correlation $^{13}$C-detected spectra of 200 μM $^{13}$C,$^{15}$N H-NS$_{ST}$Ct (orange) and 150 μM $^{13}$C,$^{15}$N H-NS$_{ST}$Ct saturated 1:10 (molar) with unlabeled 1.5 mM H-NS$_{1-57}$ (dark grey). Given a $K_d$ of ~4 μM (*Shahul Hameed et al., 2019*) over 99% of H-NS$_{ST}$Ct are expected to be in the complexed form under these conditions. The OX axis holds all of the $^{13}$C,$^{15}$N H-NS$_{ST}$Ct backbone C' carbonyl chemical shifts correlated with OY (marked $^{13}$Cab) where each amino acid stripe crosses with its own Cα and Cβ (linked by thin dotted lines). (B) Top panel: The $^{13}$C chemical shift differences as a function of residue number of H-NS$_{1-57}$ saturated $^{13}$C,$^{15}$N H-NS$_{ST}$Ct and *apo* form. The most marked changes occur in residues K89, R90, A91, and A92 that predominantly form a β-turn type VIII. (C) Structural model of the H-NS$_{ST}$Ct in transparent surface representation revealing the backbone as ribbon. The structure is based on PDB ID 2L93, but extended N-terminally in random conformation to represent the full sequence of our construct in its *apo* form. All residues experiencing significant $^{13}$C chemical shift changes upon binding to H-NS$_{1-57}$ are marked in magenta on the structure of the H-NS$_{ST}$Ct; positive residues (R+K) are labeled in bold. Bottom panel: The sequence alignment of the four selected HN-S orthologs highlighting conserved positively charged linker residues (bold) and residues implicated in binding to site1 (green). (D) Secondary structure motifs (red: helix; blue: β-sheet; green: β-turn, present only in the complex) and the RCI-$S^2$ order parameter (describing the backbone dynamics) of ligand-free (orange) and saturated (gray) H-NS$_{ST}$Ct are shown.

The online version of this article includes the following figure supplement(s) for figure 4:

**Figure supplement 1.** The low-γ $^{13}$C-detected experiments unveil the molecular details of highly dynamic and solvent-exposed residues elusive for classical $^1$H-detected approaches.

DNAbd. The center of this linker region, residues 89–92, rigidified upon binding and predominantly formed a β-turn conformation. However, the resulting intramolecular interaction was maintained through 'fuzzy' charge-pairing that did not fix the partners into a structurally stable complex.

## Autoinhibition varies among H-NS orthologs

Having established the detailed autoinhibitory interactions between site1 and the Ct region in H-NS$_{ST}$, we next examined the H-NS orthologs. Based on our structural models (initial homology models and models from conventional MD), the electrostatic surface of the Ct was well conserved across all H-NS orthologs (*Figure 5A*). This level of conservation was expected, given that this region is also required for DNA association (*Gao et al., 2017*) – a role that needs to be conserved in all H-NS. Conversely, the site1 surface that binds to Ct was not conserved across all orthologs. While

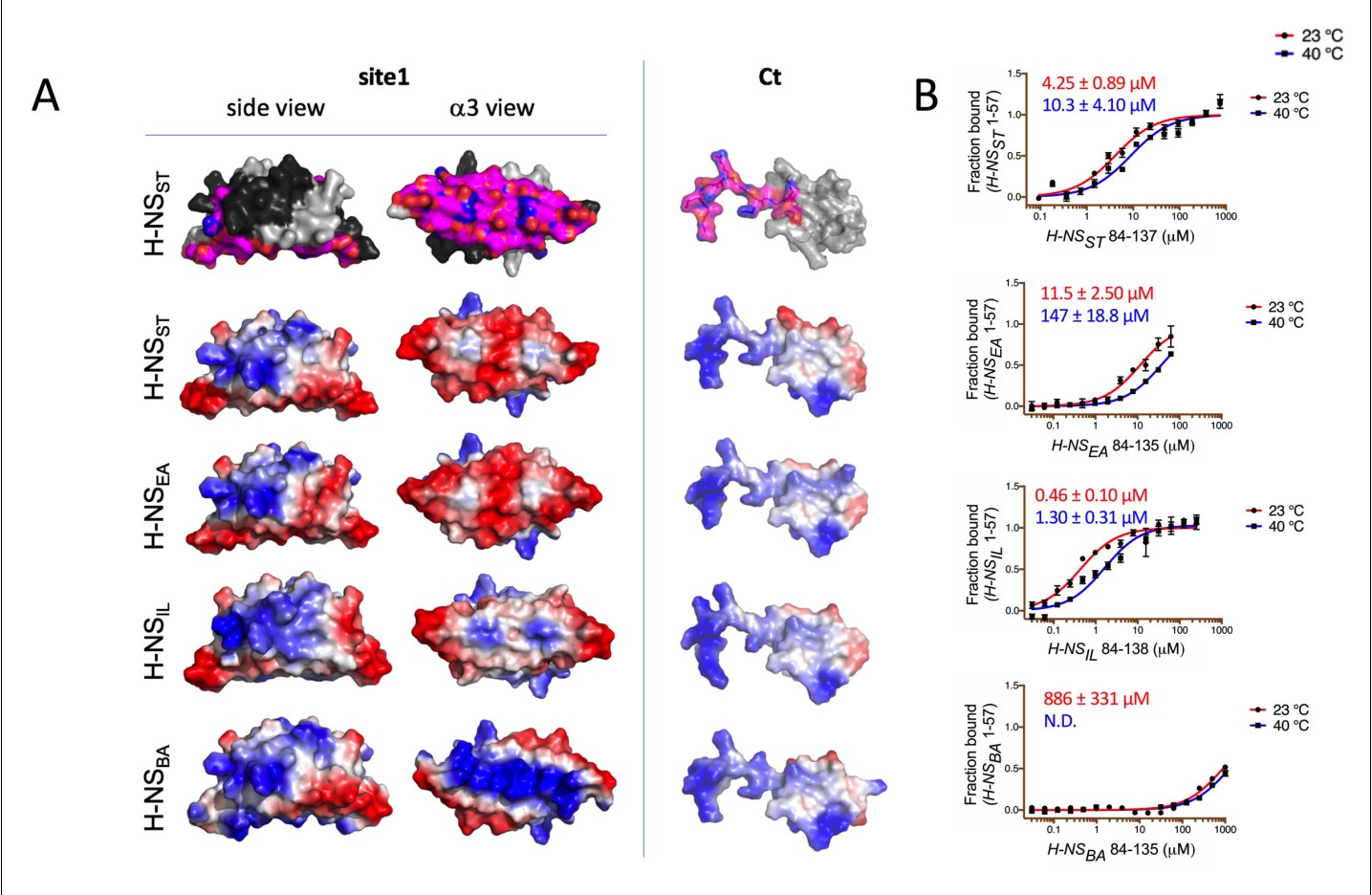

**Figure 5.** Modeling and in vitro analysis of autoinhibition and multimerization of histone-like nucleoid-structuring (H-NS) orthologs. (**A**) Left and middle panels: Surface representation of site1 of H-NS orthologs (modeled on *E. coli* H-NS, PDB accession 1NI8) shown as side and α3 (bottom) view. Right panel: Ct, comprising the linker-DNAbd fragment, residues 84–137. The top row shows the residues mapped by NMR involved in forming the autoinhibitory closed conformation (*Shahul Hameed et al., 2019* and this study). Other rows show the electrostatic surfaces, color-ramped from blue (positive) to red (negative) (calculated and visualized by Pymol). (**B**) Microscale thermophoresis (MST) titrations of unlabeled linker-DNAbd fragment onto 50 nM of Alexa-488-labeled H-NS 1-57 at 23°C (red) and 40°C (blue). The dissociation constant $K_d$ is color-coded in red (23°C) and blue (40°C). N. D.: not determined.

The online version of this article includes the following figure supplement(s) for figure 5:

**Figure supplement 1.** H-NS$_{ST}$ C-term binding to H-NS N-term.

H-NS$_{EA}$ was similar to H-NS$_{ST}$ in the overall charge distribution, α3 of H-NS$_{BA}$ showed a distinctly basic surface. H-NS$_{IL}$ displayed an intermediate electrostatic character, with features closer to H-NS$_{ST}$/H-NS$_{EA}$ (*Figure 5A*). These findings suggested that the stability of the closed conformation varies across orthologs. To test this prediction, we carried out in vitro binding experiments using microscale thermophoresis (MST).

These binding experiments between site1 and Ct confirmed that the strength of the autoassociation was similar for H-NS$_{ST}$ and H-NS$_{EA}$ (*Figure 5B*). The autoassociation was 10-fold stronger in H-NS$_{IL}$ than in H-NS$_{ST}$, despite a less acidic site1. Hence, autoinhibition in H-NS$_{IL}$ might include additional and/or different interactions. Conversely, H-NS$_{BA}$ did not show a significant capacity for autoassociation, as expected from its markedly more basic site1 surface. In agreement, the double H-NS$_{ST}$ site1 mutant E34K/E42K (designed to make the electrostatic site1 surface of H-NS$_{ST}$ *B. aphidicola* like) dramatically lowered its affinity for H-NS$_{ST}$Ct (*Figure 5—figure supplement 1*). In addition to the reduced electrostatic complementarity, H-NS$_{BA}$Ct also has a proline residue (P91) in position 3 of the β-turn region, which is highly unfavorable for this secondary structure element

(*Creighton, 1990*). Indeed, at room temperature, H-NS$_{BA}$Ct associated only very weakly with H-NS$_{ST}$ site1 ($K_d >$ mM), whereas the Ct of H-NS$_{EA}$ and H-NS$_{IL}$ bound to H-NS$_{ST}$ site1 with a similar affinity than H-NS$_{ST}$Ct ($K_d$s were 31.2 ± 3 µM and 18.2 ± 2 µM, respectively *Figure 5—figure supplement 1A,B*). Given that all four Cts display comparable electrostatic surfaces, the loss of affinity for H-NS$_{BA}$Ct supported the importance of the β-turn. Increasing the temperature decreased the self-association strength 2- to 3-fold in H-NS$_{ST}$ and H-NS$_{IL}$ and more than 10-fold in H-NS$_{EA}$ (*Figure 5B*). It also decreased the $K_d$ for H-NS$_{BA}$ to values beyond the measurement range.

We concluded that the strength of the autoinhibitory conformation is mostly modulated by the electrostatic surface characteristics of site1. The Ct is constrained by the requirement to preserve the overlapping DNA binding surface, but can decrease autoinhibition by disrupting the β-turn conformation.

## H-NS orthologs show adaptive features in vitro

We next experimentally assessed the response of the H-NS orthologs to physicochemical changes using dynamic light scattering (DLS). DLS provides the average hydration radius $R_H$ of the particles in solution, and hence gives a proxy for the tendency of H-NS molecules to form site2-mediated multimers or (still site1-linked) dimers. Thus, the $R_H$ is a convoluted signal of both effects, that is the relative strength of site2 multimerization and of the autoinhibitory conformation (if it exists). We measured the $R_H$ under different salt concentrations and temperatures. As reported previously, H-NS$_{ST}$ showed a clear drop in $R_H$ from 10°C to 40 °C (*Figure 6A*; *Shahul Hameed et al., 2019*). The marked decrease of $R_H$ for curves at 0.15, 0.25, and 0.50 M NaCl indicated a strong inverse

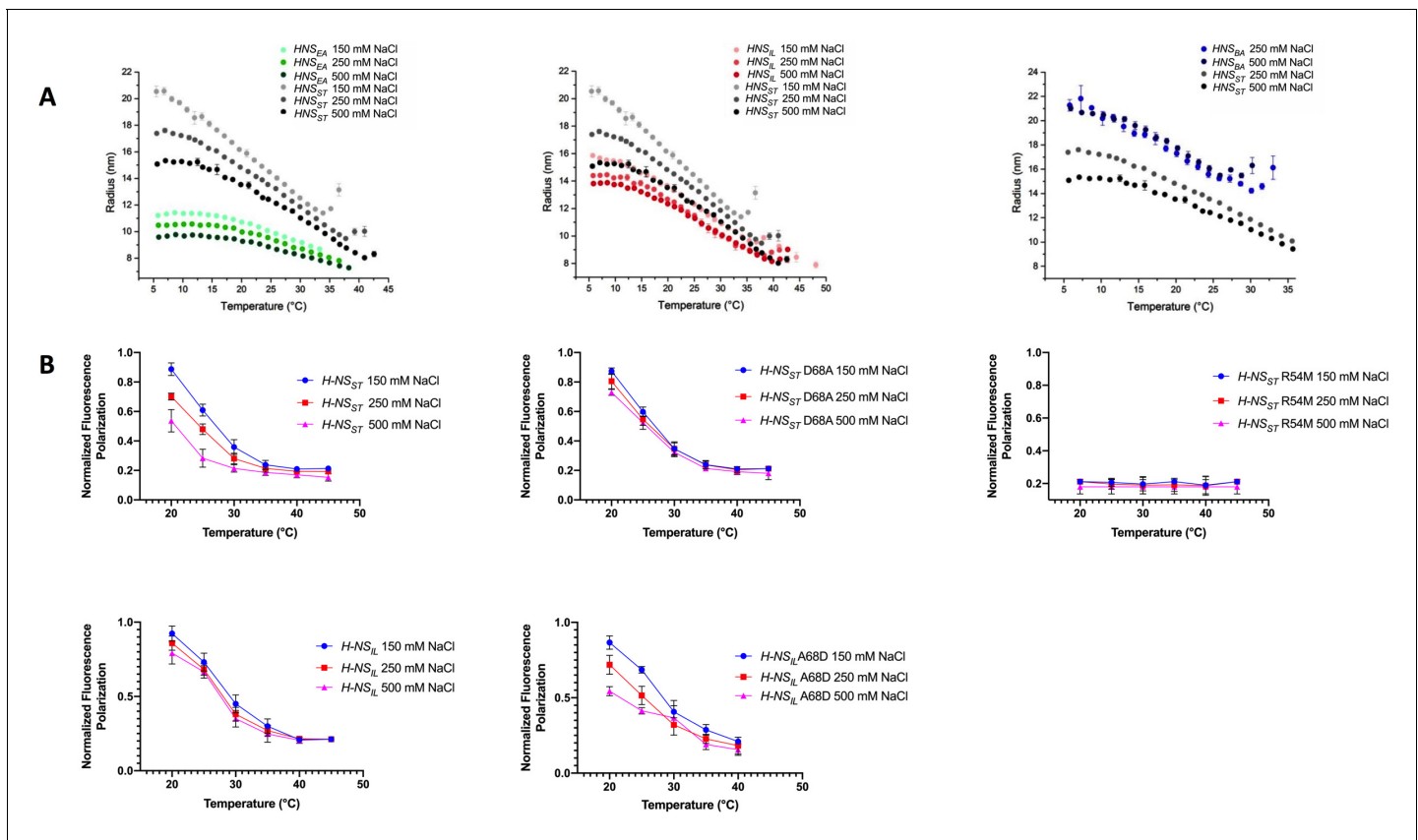

**Figure 6.** Temperature- and salt-dependent oligomerization of histone-like nucleoid-structuring (H-NS) proteins. (A) Dynamic light scattering (DLS) experiments showing changes in hydrodynamic radius (as a proxy of apparent site2 stability) upon changes in salinity and temperature. Data in (B, D) are means ± S.D., n = 3. (B) Fluorescence anisotropy to determine oligomerization of H-NS site1–site2 (residues 1–83) and mutants at various salt concentrations. Unlabeled H-NS$_{ST}$ (1–83) was titrated against Alexa-488-labeled H-NS$_{ST}$ (1–83), D68A, and R54M mutants. Unlabeled H-NS$_{IL}$ (1–83) was titrated against Alexa-488-labeled H-NS$_{IL}$ (1–83) and A68D mutant.

correlation between salinity and site2 stability, in agreement with a key role of salt bridges in stabilizing the site2 dimers.

All three H-NS orthologs displayed a similar behavior overall, further supporting that the general mechanism of site2-mediated multimerization and environment-sensing was preserved. However, we noted important differences in the orthologs' response characteristics (*Figure 6A*): (1) Of the four orthologs, H-NS$_{ST}$ responded most strongly to salinity and temperature, consistent with the broken R54-E74' salt bridge and large site2 RMSF in our high-salinity or high-temperature simulations. (2) H-NS$_{EA}$ was less temperature sensitive and showed weaker multimerization than the other orthologs. Indeed, our simulations suggested that H-NS$_{EA}$ site2 can rearrange the inter-dimer salt bridge and form either R54-E74' or R54-D71' to maintain site2 stability at higher temperatures. (3) H-NS$_{BA}$ had the highest tendency to multimerize among all the orthologs tested, which might partly be explained by the absence of the autoinhibitory conformation. Compared to H-NS$_{ST}$, our PMF calculations showed a slightly higher sensitivity to temperature and a slightly reduced sensitivity to salinity. Although these tendencies were apparent in our DLS data, these data were also affected by the fact that H-NS$_{BA}$ required more than 150 mM NaCl to stay in solution, but H-NS$_{BA}$ nonetheless aggregated at 30°C. (4) H-NS$_{IL}$ showed a decreased sensitivity to salinity compared to H-NS$_{ST}$, as suggested by our computational analysis (i.e. the lack of the site2 K57-D68' salt bridge, the lack of salt-promoted free-energy changes, and the attenuated electrostatic site1 surface).

To corroborate these conclusions, we designed several site2 mutants and tested their effect on protein multimerization using fluorescence anisotropy. To eliminate the influence of the autoinhibitory site1:Ct interaction, we used H-NS constructs that lacked the Ct (H-NS 1–83). The normalized fluorescence polarization (NFP) of H-NS$_{ST}$ declined between 20°C and ~40°C to a value of 0.2, indicating that the average particle size decreased with temperature, as observed in DLS. And as in DLS, increases in salt lowered the NFP, and hence, the propensity of the particles to form site2-linked multimers (*Figure 6B*). The H-NS$_{ST}$ R54M mutant, disrupting the 'housekeeping' salt bridges R54-E74' and R54-D71', displayed an NFP of 0.2 at all temperatures and salt concentrations. Thus, this mutant supported the key roles of the R54-mediated salt bridges in H-NS multimerization and environment-sensing. To experimentally assess the role of the K57-D68' salt bridge in conveying salt sensitivity, we introduced the D68A 'IL-like' mutation in H-NS$_{ST}$, and the A68D 'ST-like' mutation in H-NS$_{IL}$. These mutations abrogated salt sensitivity in H-NS$_{ST}$ and introduced salt sensitivity in H-NS$_{IL}$, as predicted by our computational analysis (*Figure 6B*).

Collectively, our experimental observations revealed significant differences in response to physicochemical parameters, which were in agreement with our predictions based on the molecular features of the H-NS orthologs.

## Conclusion

Environment-sensing through the pleiotropic gene regulator H-NS helps *S. typhimurium* to adapt when it is present inside its host mammal. In a previous study, we had shown that an increase in temperature, and to some extent salinity, dissociates the second dimerization element (site2). Melting of site2 produces two effects: first, it impedes synergistic DNA binding of H-NS multimers, and second, it allows H-NS to adopt an autoinhibitory conformation where DNA binding residues on the C-terminal linker-DNAbd fragment (herein abbreviated as the Ct) associate with the N-terminal site1 dimerization domain (*Shahul Hameed et al., 2019*). In our current study, we confirmed that site2 is the element that senses changes in physicochemical parameters, and we uncovered additional aspects of this process. In particular, proton-less NMR fully revealed the position and dynamics of the Ct residues involved in the autoinhibitory association with site1. We also showed that the formation of a β-turn in the linker residues 89–91 is associated with the autoinhibited conformation. The Ct residues critical for autoinhibition cannot reach site1 without site2 dissociation (*Figure 3—figure supplement 2B*), confirming that the closed autoinhibited conformation is mutually exclusive with H-NS multimerization. Our NMR analysis also demonstrated that this autoinhibition is achieved at a low entropic cost, maintaining a high flexibility with respect to the exact distribution of the interacting charges on both site1 and the linker-DNAbd fragment. On the one hand, avoiding the entropic penalty helps the autoinhibitory interaction to prevail against the competing DNA association. (Of note, the covalent link between site1 and the Ct will enhance their local concentration and hence their apparent affinity compared to our measurement based on separate domains in *Figure 5B*.) On the other

hand, the fuzziness of the charge–charge interactions facilitates preserving the capacity for autoinhibition during bacterial evolution and adaptation.

Based on our refined molecular understanding of *S. typhimurium* H-NS, we then investigated environment-sensing of H-NS orthologs from bacteria that infect plants, bacteria that are endosymbionts of insects, and bacteria that are presumably free-living in or close to a hydrothermal vent. Across all four orthologs, we observed a conceptually similar response to temperature and salt, both overall and on an atomic level, where salt bridges play key roles. This similarity suggests that environment-sensing in H-NS evolved by co-opting an ancestral feature, namely the relative instability of the simple site2 helix-turn-helix dimerization motif. However, marked idiosyncrasies in the response of H-NS orthologs suggest that this ancestral feature was then adapted to fit the current habitat and lifestyle. Thus, our analysis suggests that environment-sensing by H-NS originated from an exaptation followed by adaptation. Our combined computational and experimental structural analysis allowed us to relate the observed in vitro features of this adaptation to events on a residual level: in particular, the salt bridge disposition and stability of site2, and the strength of the autoinhibition governed mostly by the electrostatics of site1 helix α3.

Although other factors inside bacteria can modify the in vitro behavior of the isolated protein, it is interesting to consider these idiosyncrasies with respect to the bacteria's habitats (*Figure 7*):

1. H-NS$_{ST}$ had the highest sensitivity to temperature and salt, in agreement with the critical role of H-NS$_{ST}$ in helping Salmonella adapt its gene expression profile depending on if it is inside or outside a warm-blooded mammal.

2. In comparison, we found that the response to temperature was markedly attenuated in H-NS$_{EA}$. *E. amylovora* is the causing agent of fire blight, a contagious disease that mostly affects apples and pears (*Vrancken et al., 2013*). The reduced sensitivity of H-NS to temperature may reflect the minor importance of this factor in an environment of ambient temperature in temperate climate zones.

3. *B. aphidicola* is an intracellular symbiont of aphids that is maternally transmitted to the next generation *via* the ovaries (*Douglas, 1998*). *B. aphidicola* co-evolved with aphids for more

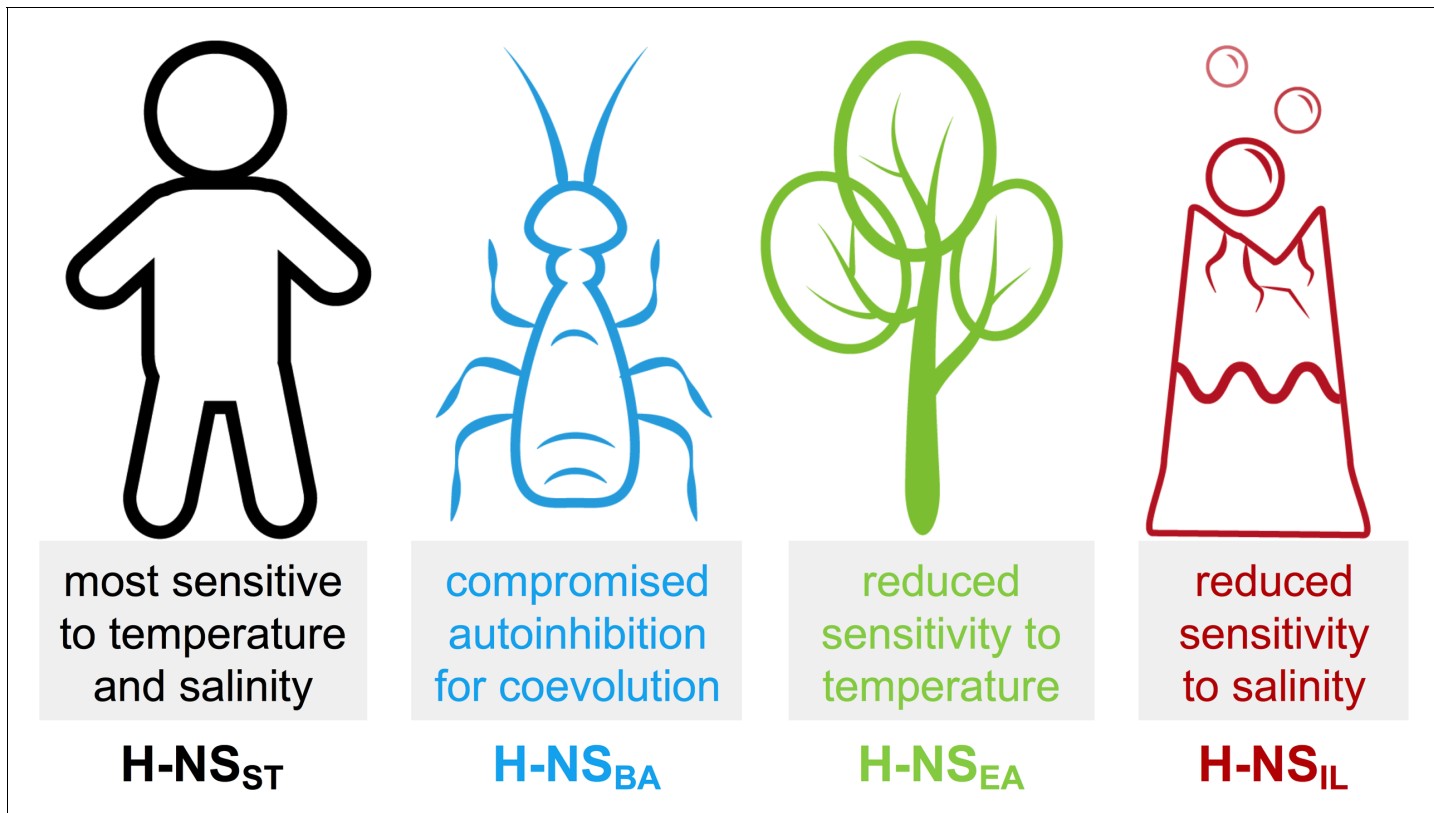

**Figure 7.** Summary of the most notable adaptations in environment-sensing observed for the histone-like nucleoid-structuring (H-NS) orthologs.

than 150 million years, and despite having the highest sequence identity (61%) to H-NS$_{ST}$ of all orthologs, H-NS$_{BA}$ showed the least conserved features among the orthologs tested, indicating that adaptive evolution was achieved by only minor changes. H-NS$_{BA}$ site2 interactions were stronger than those of other orthologs, and the features promoting the autoinhibitory form were compromised. Hence, H-NS$_{BA}$ may provide a stronger and more robust repression of the genes that it controls. In vitro, H-NS$_{BA}$ was the least stable ortholog tested and had already started to aggregate above 30℃, in agreement with the fact that *B. aphidicola* cannot survive temperatures of 35℃ for extended periods.

4. Despite having a sequence identity least similar to H-NS$_{ST}$ (41%), H-NS$_{IL}$ maintained an overall similar response profile. However, with an aggregation temperature of 45–50℃, H-NS$_{IL}$ was the most heat stable, especially at low pH and high salinity, as expected for a thermophilic and halophilic bacterium. Moreover, autoinhibition was 10-fold stronger than in Salmonella H-NS and relatively little affected by heat. The natural environment of *I. loihiensis* (hot hydrothermal fluids venting into cold seawater) provides a temperature range from 4℃ to 163℃ (*Donachie et al., 2003*), suggesting that temperature-sensing by H-NS$_{IL}$ is biologically relevant. The attenuated response of H-NS$_{IL}$ to salinity might reflect the capacity of *I. loihiensis* to grow in 20% (wt/vol) NaCl medium.

Our integrative approach provided atomistic insights on how residue-level substitutions on a protein support adaptation of organisms to different lifestyles.

## Materials and methods

Using the CHARMM36 all-atom force field, we performed conventional MD simulations of H-NS tetramers and simulations with US enhanced sampling of site2 dimers in GROMACS. For DLS and MST, recombinant protein production and measurements were adapted from *Shahul Hameed et al., 2019*. However, we fluorescently labeled site1 for DLS, instead of the Ct. For NMR, $^{13}$C,$^{15}$N-labeled *S. typhimurium* H-NS$_{84-137}$ was expressed in minimal M9 media with 5 g/L of [U-13C] glucose and 1 g/L of $^{15}$NH$_4$Cl salt. Proton and low-γ detected high-resolution NMR spectroscopy was carried out on a 700MHz Bruker Avance NEO spectrometer equipped with a 5 mm TXO cryogenic probe optimised for 15N and 13C direct detection at 25℃. Details are shown below.

### Computational methods

#### Model preparation

We built our homology model of full-length H-NS (UniprotID: P0A1S2) based on orthologs in Swiss-Model (*Arnold et al., 2006*) with the templates for the dimerization domain (PDB ID: 3NR7) and the DNA-binding domain (PDB ID: 2L93). The site2 dimer models were initiated in an anti-parallel configuration, while the tetramer models were constructed according to the crystal packing (PDB ID: 3NR7). Maestro (Schrödinger, Inc) was used to construct the full-length model from different domains.

#### Simulation setup

Our simulations were carried out by GROMACS (*Wassenaar et al., 2013*) (MD simulations of tetramers and PMF simulations of site2 dimers). All the models were solvated in a TIP3P water box, with counterions to neutralize the charges and additional NaCl for the desired salinity. Each tetramer system contains ~33,000 TIP3P water molecules, counter ions, and 150 or 500 mM NaCl, totaling ca. 100,000 atoms in a periodic box 13 × 9 × 9 nm$^3$. All simulations were performed following a minimization, 250 ps equilibration in the NVT and NPT ensemble with Berendsen temperature and pressure coupling, and a production stage NPT (20 or 40℃, 1 bar). The CHARMM36 force field (*Best et al., 2012*) was used with the cmap correction. The particle mesh Ewald (PME) technique (*Darden et al., 1993*) was used for the electrostatic calculations. The van der Waals and short-range electrostatics were cut-off at 12.0 Å with switch at 10 Å.

The PMF simulations were carried out with the MD program GROMACS (*Wassenaar et al., 2013*) using the umbrella sampling technique. The CHARMM36 force field was also used. Each site2 monomer of the center of mass (COM) distance was chosen as the dissociation pathway and used for enhanced sampling. After 500 ps equilibrium with the NPT ensemble, initial structures for windows along the reaction coordinates were generated with steered MD. In the steered MD

simulation, one chain was pulled away along the direction of increasing the COM distance with a force constant of $k$ = 12 kcal mol$^{-1}$ Å$^{-2}$, until the COM distance reached 25 Å. The windows were taken within a range of 0–25 Å. The umbrella windows were optimized at the 0.3 Å interval to ensure sufficient overlap. There are about 80 windows per simulation, and each window was simulated with a force constant of 1.2 kcal mol$^{-1}$ Å$^{-2}$. All PMF simulations converged in 54 ns per window (*Figure 3—figure supplement 2A*). The helicity percentage of initial and final structures was measured for each window. For all windows, the helicity percentage was approximately 87.5% for the initial models and 85% for the final ones (*Figure 3—figure supplement 2B*).

## Experimental methods

### Protein production

*S. typhimurium* H-NS$_{1-57,C21S}$, H-NS$_{1-57,C21S,E34K,E42K}$, H-NS$_{1-83, C21S}$, H-NS$_{1-83,C21S,D68A}$, H-NS$_{1-83, C21S,R54F}$, H-NS$_{84-137}$, and H-NS$_{84-137,K89E,R90E}$ were cloned and produced as described previously (*Shahul Hameed et al., 2019*). *E. amylovora* (H-NS$_{1-57}$, H-NS$_{82-134}$), *B. aphidicola* (H-NS$_{1-57}$, H-NS$_{84-135}$), and *I. loihiensis* (H-NS$_{1-57}$, H-NS$_{1-83}$, H-NS$_{1-83,A68D}$, and H-NS$_{85-138}$) genes were individually cloned into pGEX6P-1, expressed, and purified as described previously (*Shahul Hameed et al., 2019*). For the high-resolution nuclear magnetic resonance (NMR) studies, the uniformly double $^{13}$C,$^{15}$N-labeled *S. typhimurium* H-NS$_{84-137}$ (with additional GPLG residues before S$^{84}$) was expressed in minimal M9 media with 5 g/L of [U-$^{13}$C] glucose and 1 g/L of $^{15}$NH$_4$Cl salt. The unlabeled *S. typhimurium* H-NS$_{1-57}$ N-terminal domain was expressed and purified as described before (*Shahul Hameed et al., 2019*). The final NMR buffer was 50 mM NaCl, 2% (vol/vol) D$_2$O, 20 mM Bis–Tris at pH 6.5%, and 0.002% NaN$_3$.

### Dynamic light scattering

For DLS measurements, H-NS from *S. typhimurium*, *E. amylovora*, *B. aphidicola*, and *I. loihiensis* were expressed as N-terminal mCherry fusion proteins with an N-term His tag in *E. coli* BL21 using the expression vector pET28b. The linker sequence SAGGSASGASG was inserted between mCherry and H-NS proteins to avoid steric clashes in the dimer. Bacteria were grown in LB medium, induced with 1 mM IPTG at 25℃ overnight. Cells were harvested and resuspended in lysis buffer (50 mM Tris pH8, 500 mM NaCl, 10 mM Imidazole with addition of lysozyme, DNase I, and 1% triton X-100) and lysed by mild sonication. Proteins and bacterial membranes were separated by centrifugation (30 min, at 15,000 × *g*), and the supernatant was applied to Ni-NTA beads (Qiagen) for 2 hr. The column was washed thoroughly with 50 mM Tris pH8, 500 mM NaCl, 10 mM Imidazole, and protein was then eluted with 50 mM Tris pH8, 500 mM NaCl, 400 mM imidazole, and 1 mM dithiothreitol. After dialysis in 50 mM HEPES pH7.4, 300 mM NaCl, 0.5 mM TCEP, eluted protein was further purified by ion-exchange chromatography using either MonoQ or MonoS column (GE) in the same buffer. Protein multimerization was observed in the combination of different salt (150, 250, and 500 M NaCl) and pH (6, 7, and 8) conditions. For this, 100 mM MES, MOPS, and HEPES buffers were used, with proteins at concentrations ranging from 125 to 500 µM, in a final volume of 100 µL. Dynamic light scattering measurements were performed in 96-well plates (Greiner) using a DynaPro plate reader-II (Wyatt Technologies). A triplicate of three wells was measured for every sample with five acquisitions of 5 s for every well. The machine was cooled with gaseous nitrogen, with a starting temperature of 5℃, followed by an increase to 60℃ at a ramp rate set so that each well is measured every 1℃. Data were analyzed with DYNAMICS software (Wyatt Technologies) as temperature dependence and exported for further fitting on Origin software using a Logistic Fit. The presented results are mean values with standard error mean determined from the triplicate sample.

### Proton and low-γ detected high-resolution NMR spectroscopy

All NMR measurements were done on 700 MHz Bruker Avance NEO spectrometer equipped with a 5 mm cryogenic TXO direct detection probe optimized for $^{15}$N and $^{13}$C direct detection at 25℃. The sequence-specific backbone resonance assignments of visible H/N correlations on $^1$H-detected spectra of *S. typhimurium* H-NS$_{84-137}$ protein at 200 µM concentration in *Apo* and H-NS$_{1-57}$-saturated (1.5 mM) forms were achieved with classical set of triple-resonance experiments, that is HNCA, HncoCA, HNCO, HNcaCO, HNCACB, CBCAcoNH (*Sattler, 1999*) and previously published assignments (*Shindo et al., 1995*). The 100% complete sets of Cα, Cβ, and C' resonances for *Apo*

and H-NS$_{1-57}$-saturated (1.5 mM) forms covering the entire protein sequence, together with the residues not visible on H/N correlation $^1$H-detected experiments (due to amide exchange with water), were achieved with intra-residue 2D (H)CACO (*c_hcaco_ia3d*, 16 scans) and (H)CACBCO (*c_hcbcaco_ia3d*, 32 scans) supported with sequential (H)CANCO (*c_hcanco_ia3d*, 96 scans) $^{13}$C-detected experiments (*Gray et al., 2012*). The low-γ, $^{13}$C-detected experiments mentioned above were started with $^1$H-excitation in order to enhance the sensitivity and recorded in in-phase and anti-phase mode for the virtual decoupling. All spectra were processed in NMRpipe and analyzed in CARA and Sparky software. The random-coil-index order parameters RCI-$S^2$ and secondary motifs, like β-turn, for *Apo* and H-NS$_{1-57}$-saturated (1.5 mM) forms were determined from complete lists of C$_\alpha$, C$_\beta$ (except glycines), N, C' chemical shifts with the TalosN and MICS programs, respectively. The significance of chemical shift perturbations (CSPs) was established as follows: We calculated the combined CSP of backbone $^{13}$C atoms, that is CSP = sqrt($\Delta\sigma_{C\alpha}$2 + $\Delta\sigma_{CO}$2), and selected the residues that are above the median+1.5*IQR (interquartile range) as the cut-off (1.5*IQR corresponds to ~2.7*S.D.)

## MST for protein–protein interactions

H-NS (residues 1–57) from *S. typhimurium* and its double mutant (E34K, E42K), *E. amylovora*, *B. aphidicola*, and *I. loihiensis* were individually labeled N-terminally with fluorescent Alexa-488-TFP (Thermo Scientific) and then unlabeled C-term of those proteins and *S. typhimurium* double mutant K89E, R90E were titrated against Alexa-488-labeled N-term correspondingly and the final results were plotted as described previously (*Shahul Hameed et al., 2019*).

## Fluorescence anisotropy to determine protein oligomerization

H-NS$_{ST\ 1-83,C21S}$ and H-NS$_{IL\ 1-83}$ were N-terminally labeled with Alexa-488-TFP (Thermo Scientific) (*Shahul Hameed et al., 2019*). 200 μM of unlabeled H-NS$_{ST\ 1-83,C21S}$, H-NS$_{ST\ 1-83,C21S,D68A}$, H-NS$_{ST\ 1-83,C21S,R54F}$, H-NS$_{IL\ 1-83}$, and H-NS$_{IL\ 1-83,\ A68D}$ were added to 1 μM of the corresponding labeled H-NS proteins with the final volume of 25 μL. The proteins were incubated for 30 min, and then measurements were recorded at a temperature ranging from 20℃ to 45℃ with intervals of 5℃. Measurements were using black/clear 384 well plate (Corning) with PHERAstar FS microplate reader (BMG Labtech) installed with a fluorescence polarization filter. The excitation wavelength was 480 nm, and emission was 520 nm. Polarization with a gain of 40% was used to measure the initial fluorescence polarization of 1 μM protein. Polarization was calculated using MARS data analysis software (*Pastor-Flores et al., 2020*).

# Acknowledgements

Research by UH, VK, FH, AK, ML, LJ, and SA reported in this work was supported by the King Abdullah University of Science and Technology (KAUST) through the baseline fund and the Award No FCC/1/1976–25 from the Office of Sponsored Research (OSR). CL and JMR were partially supported by the National Institutes of Health award (R01GM129431 to JL). XZ and JL were partially supported by the National Science Foundation (CAREER CHE-1945394 to JL). We acknowledge support from the KAUST Bioscience and Imaging core laboratories and the computational resources from the Vermont Advanced Compute Core (VACC) and the Anton supercomputer in Pittsburgh Supercomputing Center (PSC), and thank M Cusack (KAUST Research Support Services) for editorial help.

# Additional information

### Funding

| Funder | Grant reference number | Author |
| --- | --- | --- |
| King Abdullah University of Science and Technology | FCC/1/1976-25 | Stefan T Arold |
| National Institutes of Health | R01GM129431 | Jianing Li |
| King Abdullah University of Science and Technology | FCC/1/1976-21 | Stefan T Arold |

| National Science Foundation | CAREER CHE-1945394 | Jianing Li Xiaochuan Zhao |

The funders had no role in study design, data collection and interpretation, or the decision to submit the work for publication.

## Author contributions

Xiaochuan Zhao, Data curation, Software, Formal analysis, Investigation, Writing - original draft; Umar F Shahul Hameed, Conceptualization, Data curation, Formal analysis, Validation, Investigation, Methodology, Writing - original draft, Writing - review and editing; Vladlena Kharchenko, Formal analysis, Investigation, Methodology; Chenyi Liao, Software, Formal analysis; Franceline Huser, Anand K Radhakrishnan, Conceptualization, Formal analysis, Investigation, Methodology; Jacob M Remington, Investigation, Visualization, Writing - review and editing; Mariusz Jaremko, Conceptualization, Data curation, Supervision, Validation, Investigation, Methodology, Writing - original draft, Writing - review and editing; Łukasz Jaremko, Conceptualization, Data curation, Formal analysis, Supervision, Validation, Investigation, Methodology, Writing - original draft, Project administration, Writing, review and editing; Stefan T Arold, Conceptualization, Formal analysis, Supervision, Funding acquisition, Investigation, Methodology, Writing - original draft, Project administration, Writing - review and editing; Jianing Li, Conceptualization, Resources, Supervision, Funding acquisition, Investigation, Visualization, Methodology, Writing - original draft, Project administration, Writing - review and editing

## Author ORCIDs

Xiaochuan Zhao (iD) https://orcid.org/0000-0002-0127-4789
Umar F Shahul Hameed (iD) https://orcid.org/0000-0002-0552-7149
Łukasz Jaremko (iD) https://orcid.org/0000-0001-7684-9359
Stefan T Arold (iD) https://orcid.org/0000-0001-5278-0668
Jianing Li (iD) https://orcid.org/0000-0002-0143-8894

## Decision letter and Author response

Decision letter https://doi.org/10.7554/eLife.57467.sa1
Author response https://doi.org/10.7554/eLife.57467.sa2

# Additional files

## Supplementary files

• Supplementary file 1. (**A**) The sequence similarity matrix of H-NS$_{ST}$, H-NS$_{EA}$, H-NS$_{BA}$, and H-NS$_{IL}$. (**B**) Summary of reported H-NS simulations (MD = unbiased molecular dynamics simulation; US = umbrella sampling simulations). The CHARMM36 force field (*Williams and Rimsky, 1997*) with TIP3P water model was used. Total simulation length = 5.7 µs. (**C**) The average RMSF of each helical region in H-NS site1/site2 at different conditions. The average of the last 10 ns of a total of 200 ns of both replicas were used. (**D**) Free-energy prediction of mutations by prediction tools. (**E**) Comparison of prediction tools and free-energy calculations. (**F**) Statistics of conservative charged contacts in MD simulations (side chain N-O distances in Å, averaged over the last 50 ns of two simulation replicas).

• Transparent reporting form

## Data availability

NMR chemical shift assignments were deposited at the BMBR https://betadeposit.bmrb.wisc.edu/ with IDs 50239 and 50240.

The following datasets were generated:

| Author(s) | Year | Dataset title | Dataset URL | Database and Identifier |
|---|---|---|---|---|
| Kharchenko V, | 2020 | Chemical shifts of H-NS C-term | https://bmrb.io/data_li- | BMRB, 50239 |

| Jaremko M, Jaremko L | | with linker | brary/summary/index. php?bmrbId=50239 | |
| Kharchenko V, Jaremko M, Jaremko L | 2020 | Chemical shifts of H-NS C-term with linker against the N-term site1 | https://bmrb.io/data_li-brary/summary/index. php?bmrbId=50240 | BMRB, 50240 |

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
