## [Decision Letter]

**Acceptance summary:**

Combining simulation, NMR and biophysical experiments, this study investigated the molecular basis of the temperature- and salinity-dependence of DNA availability regulated by H-NS protein. The study compared four orthologs from human, plants, insects and deep-sea hypothermal vent and illustrated that modulation of the strength of electrostatic interactions in H-NS oligomerization underlies the environment sensing.

**Decision letter after peer review:**

Thank you for submitting your article "Molecular Basis for the Adaptive Evolution of Environment Sensing by H-NS Proteins" for consideration by *eLife*. Your article has been reviewed by four peer reviewers, including Yibing Shan as the Reviewing Editor and Reviewer #1, and the evaluation has been overseen by Cynthia Wolberger as the Senior Editor.

The reviewers have discussed the reviews with one another and the Reviewing Editor has drafted this decision to help you prepare a revised submission.

Summary:

The manuscript reports a study that combines simulation, NMR and biophysical experiments to determine the molecular basis of environment sensing of the histone-like nucleoid-structuring (H-NS) protein from enterobacteria, which regulate DNA availability for transcription in response to environmental cues including temperature and salinity. The study compared four orthologs from human, plants, insects and deep-sea hypothermal vent and concluded that the differential strength of the electrostatic interactions at the so-called site 1 of H-NS in its oligomerization may explain the different temperature and salinity-dependence of these orthologs. The authors argue that this provide a molecule mechanism for the adaptive evolution of the H-NS proteins.

1) The reviewers raised concerns that some of the key conclusions do not have direct experimental support, while the simulation observations are highly qualitative. The conclusion regarding the role of electrostatic interactions in the evolutionary adaption needs substantiation. Point mutations at Site 1 should be tried to validate this conclusion. Alchemical free-energy calculation (FEP) on mutations at selected residues at Site 1 should be considered.

2) There is not enough direct evidence to attribute the difference in site1 and Ct binding to "dynamic electrostatic interactions". At minimum, a salt dependence study should be done. Ideally, one should perform site-specific mutagenesis experiments to test some of the specific contacts discussed in detail from MD trajectories.

3) Regarding the interpretation of how site2 sequences alter sensing sensitivity. The discussion is speculative and rely solely on the presence and absence of various salt bridges during MD. Again, such detailed predictions need to be substantiated by experimental data, such as mutagenesis data.

4) Has the evolution relied on one or two specific mutations to introduce certain behavior, or relied on some more extensive combination of mutations. The authors should perform more extensive sequence comparison covering more bacteria H-NS proteins in more species.

5) A key point is the presence of a b-turn in the structure around R90 that inverts the direction of the backbone and allows the formation of differential salt bridges. The authors used heteronuclear-detected NMR pulse sequences to assign the protein resonances and the determine the structural propensity of the protein. Protonless NMR spectroscopy was necessary as the amide resonances broadened beyond detection. How did the author decide the most significant chemical shift changes? Did they use the standard deviation from the average? Is it really the case that the NMR unequivocally determined the short type VIII β-turn in residues 89-92, or that's just one possible conformation consistent with the data. The authors needs to present more details of their analysis and carefully calibrate their claim with respect to this finding. What is the binding affinity between CTst and site 1? what are the concentrations used in NMR? How do we rule out nonspecific effects of CS changes? Figure 4 legend does not have a description of panel C. What coordinates did the authors use to render the residues involved in the b-turn? What type of b-turn is represented? Figure 4 Panels B and C do not seem to match.

6) The PMF calculation provides a nice and self-consistent picture of the temperature and salt dependence of site2 dimerization. However, site2 is persumably unstable in monomer form based on the authors previous study (and the current one), but how is unfolding accounted for in PMF calculations. Has restraints to the folded structure been applied in the PMF calculation? In any case this potential needs to be discussed in the revision.

7) The claims regarding exaptation and therapeutics from the Abstract are not even discussed or illustrated.

8) DLS experiments appear somewhat low in information content. The interpretation of the DLS measurements may be complicated by the interplay of multimerization and intramolecular compaction. This needs to be discussed and calibrated. Mutatgenesis combined with DLS can help. Alternative approaches like analytical ultracentrifugation or non-equilibrium measurements such as SEC would increase the confidence in the conclusions.

---

## [Author Response]

Revisions for this paper:1) The reviewers raised concerns that some of the key conclusions do not have direct experimental support, while the simulation observations are highly qualitative. The conclusion regarding the role of electrostatic interactions in the evolutionary adaption needs substantiation. Point mutations at Site 1 should be tried to validate this conclusion. Alchemical free-energy calculation (FEP) on mutations at selected residues at Site 1 should be considered.

Firstly, we have performed SEC-MALS and DSLS experiments to corroborate our computational predictions that the N-terminal site1 of all four orthologs are stable dimers in the temperature range relevant for environment sensing (i.e. up to ~37 degC; reported in the new Figure 2—figure supplement 2). Thus, we now further substantiate that the sensing mechanism is located on site2 for the orthologs.

Next, we used site1 point mutations to further support our observations on the evolutionary regulation of the autoinhibitory interaction between the C-terminal region (Ct) and the N-terminal site1. As we explain more in detail in our response to comment 2), these results substantiate the role of electrostatic interactions and of the β-turn in modulating the site1/Ct autoinhibition. The establishment of the N/C cross-reactivity (Figure 5—figure supplement 1) and the use of the autoinhibition-incapable site1-site2 construct for additional experiments (see below and Figure 6) allow us now to separate these two events. These additional data are now included in the manuscript (Figures 6, Figure 5—figure supplement 1 and associated text).

As suggested, we have carried out more computational analyses to strengthen our observation that site1 is stabilized by hydrophobic interactions. We selected all residues on site1 involved in intramolecular interactions (L5, L8, I11, R12, L14, L23, L26, E28, L30, L33, V36, and E39). Each residue was replaced in silico by Ala, Lys, Arg, Glu, Asp, Gln, or Asn, resulting in a total of 81 mutants on site1 to be considered. We then utilized two mutant stability prediction tools, namely Maestro (BMC Bioinformatics 2015, *16*, 116) and PremPS (https://lilab.jysw.suda.edu.cn/research/PremPS/). The mutants that produced a significant destabilizing ΔΔG change of |ΔΔG|>1.0 kcal/mol in both programs were substitutions of hydrophobic residues (see Supplementary file 1D). These mutations that were most destabilizing were: L5A, L5D, L8A, L8K, L8D, L8N, L23A, and L26A. These empirical predictions allow us to focus on several mutants for more reliable free energy calculations, which were carried out with GROMACS. There is an overall agreement between the calculated ΔΔG, but there are some noticeable differences (see Supplementary file 1E). We concluded that while this type of calculation provides a general support to our conclusion about site1 stabilisation through hydrophobic interactions, these conclusions are qualitative, not quantitative. We have added these data and the following sentence to the revised manuscript: “Our in silico mutant stability prediction analysis corroborated qualitatively the importance of hydrophobic residues for stabilising the site1 dimer, in particular of L5, L8, L23 and L26 (Supplementary file 1D and 1E).”

Concerning site2, we have introduced site-specific mutations to experimentally assess the importance of the “housekeeping” salt bridges R54-E74’ and R54-D71’ and the role of the K57-D68’ salt bridge in conveying salt sensitivity. Using fluorescence anisotropy (as an orthogonal method to DLS) we could demonstrate that these experiments confirm our computational predictions. Of note, these experiments were carried out using the site1-site2 construct (residues 1-83) to avoid potential convolution with a variation of the strength of the autoinhibition. These results are given in more detail in our response to comment 3) and the data are now shown in Figure 6.

2) There is not enough direct evidence to attribute the difference in site1 and Ct binding to "dynamic electrostatic interactions". At minimum, a salt dependence study should be done. Ideally, one should perform site-specific mutagenesis experiments to test some of the specific contacts discussed in detail from MD trajectories.

We would like to point out that we have already reported the salt dependence study for H-NS_ST_ in our previous work [NAR 2019 (doi: 10.1093/nar/gky1299)], where we showed that increasing buffer salt concentration decreases the interaction between the H-NS_ST_ site1 and Ct. We therefore focused on the site-specific mutagenesis experiments, as suggested by the reviewer. We experimentally assessed our prediction that the electrostatic surface of site1 helix3 sustains the autoinhibitory conformation through charge-pairing with the Ct. We now show that the *S. typhimurium* (*ST*) double site1 mutant E34K/E42K (which makes the electrostatic surface of site1 more like the one of *B. aphidicola* [*BA*], see Figure 5—figure supplement 1C,D) dramatically lowers the N/Ct affinity (as seen in H-NS_BA_ site1/Ct interactions), confirming the importance of electrostatics (reported in the extended section “Autoinhibition varies among H-NS orthologs” and data are shown in the new Figure 5—figure supplement 1).

We also tested the cross-reactivity of the Ct domains from all three orthologues with the site1 from H-NS_ST_. As predicted, we observe that the interaction between the site1 from H-NS_ST_ with the Ct from *E. amylovora* (*EA*) or *I. loihiensis* (*IL*) is similar in strength, supporting that the differences in the site1 are the driving forces for the differential binding of site1/Ct in H-NS_ST_, H-NS_EA_ and H-NS_IL_. However, the Ct of H-NS_BA_ only shows a very weak interaction with the *ST* site1. While preserving the charge profile of the other Cts (as needed to preserve the overlapping DNA binding surface), H-NS_BA_ specifically has a β-turn–breaking proline in the position where a β-turn was observed forming in the site1/Ct interaction in H-NS_ST_. This result with the H-NS_BA_ Ct highlights the role of the β-turn in the site1/Ct association, supporting our NMR analysis (Figure 5—figure supplement 1).

Collectively, these experiments confirm our computational predictions and also support the importance of the Ct betaturn, in agreement with our (now extended) NMR analysis. These experiments are now shown in Figure 5—figure supplement 1 and discussed in the associated section.

3) Regarding the interpretation of how site2 sequences alter sensing sensitivity. The discussion is speculative and rely solely on the presence and absence of various salt bridges during MD. Again, such detailed predictions need to be substantiated by experimental data, such as mutagenesis data.

In response to this comment, we have designed a series of site2 mutants, where we have specifically targeted the electrostatic interactions identified computationally. Thus, we have introduced site-specific mutations to experimentally assess the importance of the “housekeeping” salt bridges R54-E74’ and R54-D71’ and the role of the K57-D68’ salt bridge in conveying salt sensitivity. Using fluorescence anisotropy (as an orthogonal method to DLS) on H-NS site1-site2 (without the Ct), we could demonstrate that the R54M substitution leads to the loss of site2-linked multimerization and hence environment sensing. We could also show that introduction of the *IL*-like D68A mutation in H-NS_ST_ leads to loss of salt sensitivity, whereas the introduction of the *ST*-like A68D mutation in H-NS_IL_ introduces salt sensitivity. These experiments confirmed our computational predictions and are now shown in (Figure 6B).

4) Has the evolution relied on one or two specific mutations to introduce certain behavior, or relied on some more extensive combination of mutations. The authors should perform more extensive sequence comparison covering more bacteria H-NS proteins in more species.

This is an interesting question. Unfortunately, we might not have enough data to propose an answer. In our work, we wanted wanted to investigate how the H-NS environment sensing mechanism has been adapted to non-mammalian hosts or a free-living lifestyle. Therefore, we searched for bona fide candidate H-NS by BLAST. However, our sequence and 3D modelling analysis identified only H-NS_BA_, H-NS_EA_, and H-NS_IL_ as high-confidence H-NS proteins of bacteria with non-mammalian hosts. All other bacteria for which high-confidence H-NS genes are available appear to target mammals as well. Hence, a more extensive sequence annotation would need to be performed once enough high confidence H-NS from bacteria with diverse lifestyles are available. In Author response image 1, we show a preliminary multiple sequence alignment of our available H-NS orthologs and some representative H-NS sequences from H-NS infecting mammals. The boxed positions are those observed to have key roles in environmental sensing through ionic bonds (or absence thereof). As preliminary conclusions, we can state that many orthologues have this site2 well conserved and should function in the same manner as *S. typhimurium*. H-NS_EA_ achieves its much-lowered multimerization with only a couple of targeted mutations. H-NS_BA_ has a rather derived sequence, reflecting its adaptation to symbiosis. However, the influenza sequence also shows that relatively derived sequences act in mammals. However, experimental investigations would be needed to understand how this orthologue’s sequence affects its environmental response.

5) A key point is the presence of a b-turn in the structure around R90 that inverts the direction of the backbone and allows the formation of differential salt bridges. The authors used heteronuclear-detected NMR pulse sequences to assign the protein resonances and the determine the structural propensity of the protein. Protonless NMR spectroscopy was necessary as the amide resonances broadened beyond detection. How did the author decide the most significant chemical shift changes? Did they use the standard deviation from the average?

Thank you for this comment. Actually, we used an even more conservative approach: We calculated the combined chemical shift perturbation (CSP) of backbone ^13^C atoms, i.e. CSP = sqrt(Δσ_Cα_^2^ + Δσ_CO_^2^), and selected the residues that are above the median+1.5*IQR as the cut off (1.5*IQR is corresponding to ~2.7*st.dev.). These details are now listed in the Materials and methods section.

Is it really the case that the NMR unequivocally determined the short type VIII β-turn in residues 89-92, or that's just one possible conformation consistent with the data. The authors needs to present more details of their analysis and carefully calibrate their claim with respect to this finding.

Thank you for this comment, we fully agree. The identification of the turn comes from the assigned ^13^C chemical shifts. The confidence coefficient provided by MICS algorithm based on chemical shifts was 0.69 for type VIII of β turn, with no clear indication of any other stable motifs, only the random coil. This model where the β-turn is the dominant conformation with a minor random coil contribution is in agreement with the RCI-*S*^2^ order parameter at 0.75-0.77 levels for R^90^-A^91^. Additionally, our analysis of the cross-reactivity between the *ST* site1 and the Ct of the three orthologs further supported the importance of a β-turn at this position (see our response to comment 1 above). Nonetheless, we agree with the reviewers’ suggestion, and we have extended the information given and calibrated the relevant statements according to the reviewers’ suggestion. for example:

“Upon addition of site1, the local dynamics decreased, particularly within the stretch of four amino acids K89-R90A91-A92 (RCI-S2 > 0.6) that predominantly form the type VIII β-turn according to MICS. Nonetheless, the overall RCIS2 of the linker remained low, although experimental conditions resulted in >99% of ligand saturation of the labeled CtST, demonstrating that the association with site1 did not substantially restrict the linker’s movements (Figure 4D). “ and “The center of this linker region, residues 89–92, rigidified upon binding and predominantly formed a β-turn conformation.”

What is the binding affinity between CTst and site 1?

The K_d_ is 4 µM, as shown in Figure 5B, and in agreement with our previous report in NAR 2019 (doi: 10.1093/nar/gky1299).

What are the concentrations used in NMR?

The sample with [^13^C,^15^N] H-NS_84–137_ and unlabeled H-NS_1–57_ had the following concentrations of 150 μm and 1.5 mM, respectively. Under these conditions with the above K_d_ value we have over 99% of the labelled C-terminal domain in the complexed form. The measurement under this high level of complex strongly supports the structural (i.e. predominant β-turn) and dynamics conclusions based on the ^13^C chemical shifts.

We added this information to the text and figure legend to support our conclusions, for example, in the legend of Figure 4:

“Given a *K_d_* of ~4 µM (*17*) over 99% of H-NS_ST_Ct are expected to be in the complexed form under these conditions.”; and in the related text (already given above).

How do we rule out nonspecific effects of CS changes?

We can rule out non-specific effects based on the following: (i) The significant CSPs correspond to non-continuous protein stretches (S84-R93 and T110-G111) but cluster on a defined region on the Ct domain. (ii) We previously showed that when we deleted the region 84-91 of the Ct domain, then the interaction is lost (NAR 2019; doi: 10.1093/nar/gky1299). (iii) We now show in the revised version that the site1/Ct binding can be disrupted by sitespecific mutations on site1; (iii) we now show that the Ct domain from *B. aphidicola*, which has a preserved charge profile, but a b-turn–disrupting proline, does not bind to the *S. typhimurium* N-term domain.

Figure 4 legend does not have a description of panel C.

We apologize for this mistake, and have corrected it in the revised version.

What coordinates did the authors use to render the residues involved in the b-turn? What type of b-turn is represented?

Our displayed structural model represents an H-NS_ST_Ct *apo* form. The structure is based on PDB ID 2L93, but extended N-terminally in random conformation to represent the full sequence of our construct in an *apo* form. We now clearly state this in the legend of Figure 2C: “Structural model of the H-NS_ST_Ct in transparent surface representation revealing the backbone as ribbon. The structure is based on PDB ID 2L93, but extended N-terminally in random conformation to represent the full sequence of our construct in its apo form.”

Figure 4 Panels B and C do not seem to match.

We apologize for this oversight. Now it is corrected.

6) The PMF calculation provides a nice and self-consistent picture of the temperature and salt dependence of site2 dimerization. However, site2 is persumably unstable in monomer form based on the authors previous study (and the current one), but how is unfolding accounted for in PMF calculations. Has restraints to the folded structure been applied in the PMF calculation? In any case this potential needs to be discussed in the revision.

We agree with the reviewer that unfolding of originally folded protein may lead to inconsistency in PMF calculation.

Moreover, originally unfolded regions may become folded during the simulation, and thus influence the PMF as well. In our case, site2 consists of α-helices and has an above-average RMSF in our MD simulations and dissociates in vitro upon heating to ~40 °C. To minimize the influence of the secondary structure on the PMF calculation, we restrained the structure through increasing the force constant k (J. Chem. Theory Comput. 2019, *15*, 4). We did not observe unfolding of the secondary structure when *k* is 12 kcal/(mol Å2) in the Umbrella Sampling. To quantitatively show the conformational change of the secondary structure during the simulation, we measured the helicity percentage at the beginning and at the end of each window. We noticed that for all windows the helicity percentage is approximately 85% (87.5% for the initial models; see Author response image 2 and Figure 3—figure supplement 2). This analysis confirmed that the secondary structure of site2 remained stable during PMF calculation. These observations suggest that the naturally unfolding of site2 under physiological condition takes longer than the timescales in our simulations

To clarify our points, we have revised our manuscript so that the section 1.2 Simulation setup now also contains the following statement: “In the steered MD simulation, one chain was pulled away along in the direction of increasing the COM distance with a force constant of k = 12 kcal mol^-1^ Å^-2^, until the COM distance reached 25 Å.” and “The helicity percentage of initial and final structures was measured for each window. For all windows, the helicity percentage was approximately 87.5% for the initial models and 85% for the final ones”

**Author response image 2. sa2fig2:** The initial and final average helicity percentage of all windows in Umbrella Sampling.

7) The claims regarding exaptation and therapeutics from the Abstract are not even discussed or illustrated.

Thank you for pointing out this issue. In our initially submitted version, we had already a paragraph in the Conclusion section discussing the exaptation as a possible origin. To increase the clarity, we have now partially reworded this part to read: “Across all four orthologs, we observed a conceptually similar response to temperature and salt, both overall and on an atomic level, where salt bridges play key roles. This similarity suggests that environment-sensing in H-NS evolved by co-opting an ancestral feature, namely the relative instability of the simple site2 helix-turn-helix dimerization motif. However, marked idiosyncrasies in the response of H-NS orthologs suggest that this ancestral feature was then adapted to fit the current habitat and lifestyle. Thus, our analysis suggests that environment sensing by H-NS originated from an exaptation followed by adaptation. ”

We think that a better knowledge of the molecular basis for H-NS multimerization and autoinhibition constitutes an important input for conceiving new therapeutics (e.g., for drugs disrupting or stabilising site2 multimerisation). However, we also agree that this statement would deserve a longer discussion. Since drug design is not a major point of our manuscript, we have decided to delete this statement.

8) DLS experiments appear somewhat low in information content. The interpretation of the DLS measurements may be complicated by the interplay of multimerization and intramolecular compaction. This needs to be discussed and calibrated. Mutatgenesis combined with DLS can help. Alternative approaches like analytical ultracentrifugation or non-equilibrium measurements such as SEC would increase the confidence in the conclusions.

To address this comment, we have produced more site1 and site2 mutants and have performed more measurements to confirm the site1 stability and to separate effects on autoinhibition from effects on site2 multimerisation (already outlined in our replies to comments 1-3). Additionally, we have used fluorescence anisotropy as an orthogonal method to DLS, and we have performed SEC-MALS measurements.